# Nitrogen Use Efficiency in an Agrisilviculture System with *Gliricidia sepium* in the Cerrado Region

**DOI:** 10.3390/plants12081647

**Published:** 2023-04-14

**Authors:** Cícero Célio de Figueiredo, Túlio Nascimento Moreira, Thais Rodrigues Coser, Letícia Pereira da Silva, Gilberto Gonçalves Leite, Arminda Moreira de Carvalho, Juaci Vitória Malaquias, Robélio Leandro Marchão, Segundo Urquiaga

**Affiliations:** 1Faculty of Agronomy and Veterinary Medicine, University of Brasília, Campus Darcy Ribeiro, Brasília 70910-970, DF, Brazil; 2Embrapa Cerrados, BR-020, km 18, Planaltina 73310-970, DF, Brazil; 3Embrapa Agrobiologia, BR-465, km 7, Seropédica 23891-000, RJ, Brazil

**Keywords:** agroforestry system, integrated crop–livestock–forest system, sustainable intensification, low-carbon emission agriculture

## Abstract

Gliricidia (*Gliricidia sepium*) is a tree legume that has great potential for use in agriculture because of its multiple-use characteristics. However, there is little information in the literature about the effect of agrisilvicultural systems on nitrogen (N) cycling. This study evaluated the effect of densities of gliricidia on N cycling under an agrisilvicultural system. The treatments were composed of different densities of gliricidia: 667, 1000 and 1333 plants ha^−1^, with a fixed spacing of 5 m between the alleys. The efficiency of N use was investigated by using the ^15^N isotope tracer. In each plot, a transect perpendicular to the tree rows was established in two positions: (i) in the corn (*Zea mays*) row adjacent to the trees, and (ii) in the corn row in the center of the alley. The N fertilizer recovery efficiency ranged from 39% in the density of 667 plants ha^−1^ to 89% with 1000 plants ha^−1^. The effect of gliricidia on the N uptake by corn was higher in the central position of the alley with 1000 plants ha^−1^. The agrisilvicultural system with 1000 plants ha^−1^ was highly efficient in the recovery of mineral N, representing an excellent option for integrated production systems in tropical regions.

## 1. Introduction

Integrated agricultural and livestock production systems, with the tree component, are being widely recommended around the world, with an emphasis on tropical and subtropical latitude regions where trees can help to mitigate greenhouse gas (GHG) emissions. In Brazil, about 11.5 million ha have been established under integrated systems [1], mainly including a consortium of annual and forage crops. The introduction of tree legumes in agricultural systems that are adapted to the environmental conditions of tropical and subtropical regions is still a challenge. These trees improve the agri-livestock production systems by supplying nitrogen (N), shading and covering the soil with their cuttings and residues [2]. In this sense, *Gliricidia sepium* (Jacq) Kunth, Walpers, is a small to medium-sized tree legume, which stands out for its multiple-use characteristics [3]. Wolz and DeLucia [4], after an extensive review on alley cropping, observed that *Gliricidia sepium* (gliricidia) and *Leucaena leucocephala* are the most used species in integrated systems in tropical regions.

The adoption of forest species associated with grain and/or forage crops is of fundamental importance for the maintenance of soil organic matter (SOM) contents and quality [5,6] and soil fertility [6,7]. Several studies have demonstrated the benefits of the integration of gliricidia with grain-producing species such as corn (*Zea mays*) [2,6,8,9,10,11] or forage species in a silvipastoral system [12]. These studies showed the various benefits of introducing tree legumes to agri-livestock systems, such as increasing N contents in soil and better nutrient cycling [2,6,11]. In a silvipastoral system, under the Brazilian semi-arid region conditions, gliricidia incorporated 47 kg of N ha^−1^ year^−1^ through gliricidia litter decomposition [13]. However, the incorporation or deposition of gliricidia biomass in the soil may not meet the need for the N of crops such as corn, which in the long term can exhaust soil fertility concerning this nutrient [14].

The literature reveals that there is still a lack of information on the use of gliricidia in integrated systems, especially in the modality of crop–forest integration (forestry), where the annual crop within the alleys is intercropped with forage species [15]. In systems integrated with forest species, the recovery efficiency of N-fertilizer applied (NFRE) by annual crops may be higher compared to monocropping. In this system, the N available in deeper soil layers, which has the potential to be leached, can be absorbed by the roots of the plants before the roots of the annual crops reach this depth [3,16]. This interaction between species in the integrated system results in increased efficiency of nutrient recycling, especially N [7]. In general, the transfer of N is greater from legume species to non-legumes than the other way around. However, this transfer of N depends on the proximity of the species and the frequency of pruning in the case of tree legumes [17]. 

The NFRE depends, among other factors, on the adopted soil management practices, whether there is conventional tillage or no tillage [18], on the N fertilization rates [19,20], on the intercropped annual and perennial plants [21], on the interactions between the roots of the intercropping crops [16], and, consequently, on the cycling of N in the system [3]. In general, agroecosystems that increase biomass and SOM have higher NFRE [16,22], mainly through the greater uptake of N that would be otherwise leached [3] or lost by N oxides to the atmosphere [23,24].

One of the ways to evaluate the efficiency of N fertilization in integrated systems is through isotopic techniques [17,25]. Thus, it is possible to measure the absorption and destination of N by crops from various sources, forms, and the timing of fertilizer application from different soil management practices, and by measuring the N transfer between intercropped plants, among other aspects that may interfere with the fertilization efficiency [17]. With the use of isotopic techniques, it was demonstrated that the inclusion of forage grasses (*P. maximum* and *B. humidicola*) intercropped with corn, without the presence of a tree, absorbed only 2.9% of the ^15^N-ammonium sulfate applied, demonstrating low interference of the grasses on the uptake of N-fertilizer by corn [21]. The introduction of the tree into the integrated production system can affect the availability of N and NFRE and, consequently, the yield of crops [10,26]. The tree legume can both increase N from the biological N fixation [27] and reduce the NFRE by corn when there is competition for water and nutrients [26]. 

In this way, there are still questions about the NFRE by corn when cultivated under agrisilvicultural systems. The present work tries to answer the following questions: (i) in which position within the alleys (of 5 m) does the tree influence the deposition of residues, shading, and the supply of N? (ii) what is the influence of the abovementioned factors on N use efficiency (NUE) by the other components of the system (corn and *P. maximum*)? By answering these questions, it will be possible to find an adequate density of gliricidia plants for an ideal integrated system that brings agro-environmental benefits. Therefore, the objective of this work was to evaluate the influence of the gliricidia plant population on the efficiency use of ^15^N-enriched ammonium sulfate by corn and the forage *P. maximum* cv. Massai and their yields when cultivated in an agrisilvicultural system in the Cerrado.

## 2. Results and Discussion

### 2.1. Nitrogen Use by Corn in Agrisilvicultural System with G. sepium

The interactions between gliricidia densities and the position of corn harvested in the plot were statistically significant for N derived from mineral fertilizer (Npdf) and from soil (Npds) and for N fertilizer recovery efficiency (NFRE) determined in corn grains. On the other hand, no significant interactions were observed between gliricidia densities and the position of corn harvested for Npdf, NFRE and Npds measured in corn straw (Table 1).

At the density of 1000 plants of gliricidia ha^−1^, corn showed a higher amount of Npdf (*p* < 0.05) than at the lowest density of 667 plants ha^−1^. The highest density (1333 plants ha^−1^) provided intermediate Npdf values in corn grains, not differing from the other densities. The presence of the tree component in the system did not reduce the absorption of Npdf by the corn cultivated in the center of the plot (alley), due to lower shading in this position [28] and, possibly, lower root competition. Makumba et al. [16] measured root development from gliricidia-corn intercropping under tropical conditions in Malawi and observed higher corn root density (1.02 cm cm^−3^) compared to gliricidia (0.38 cm cm^−3^) in the top 0.0–0.4 m soil layer. However, in the sub-soil (0.4–1.0 m), beyond the corn rooting zone, gliricidia root density was 0.65 cm cm^−3^. There could be a slight competition for soil resources between corn and gliricidia in the soil uplayers, whereas in deeper layers the plants are able to reach leached and native soil nutrients and supply these nutrients by pruning [16]. The Npdf values in the middle of the alley for this study were similar to those obtained in an integrated system, without the influence of the tree component and for the same soil type [21]. Similarly, the supply of gliricidia [29] and Leucaena [30] pruning residues resulted in higher N absorption by corn in Malaysia (Typic Paleudult) and Southern Nigeria (Alfisol), respectively. Therefore, the density of 1000 plants ha^−1^ favors the absorption of N derived from fertilizer by corn, probably due to the higher gliricidia biomass generated by the plants and synergism in the absorption of N from mineral and organic sources, and this may have favored higher soil moisture by shade without affecting the luminosity for adequate corn growth. 

In general, for all of the gliricidia densities, the highest values of Npdf in corn grains were obtained in the middle line of the plot (*p* < 0.05). The proximity effect of gliricidia resulted in a lower Npdf content in corn grains, probably due to excessive shading and competition for the absorption of N-fertilizer by the plants of gliricidia. Corn plants located in the middle of the alley benefited from an environment with less shading and competition for nutrients and water, favoring the accumulation of N in the grains. In a study by Rowe et al. [26], carried out in the humid tropics of Indonesia and with a Plinthitic Kandiudult soil, they showed a lower uptake of Npdf by the corn that was adjacent to the trees due to competition in the alley system. Chirwa et al. [28], in Southern Malawi and under an Oxic Haplustalf, demonstrated the need to prune gliricidia to avoid the competitive effects that may decrease corn yield performance. 

The content of Npdf in corn straw was not influenced by the density of gliricidia plants or the position of corn plants (*p* > 0.05). The amount of Npdf in corn straw ranged from 27.98 kg ha^−1^ in the corn row adjacent to the trees at the density of 667 plants ha^−1^ to 39.93 kg ha^−1^ in the middle of the alley at 1000 plants ha^−1^ (Table 1). The results of Npdf content in corn straw had greater variation in the agrisilvicultural system, compared to an intercropped system with corn and *Panicum* sp. studied by Coser et al. [21], possibly due to the effect of interception solar radiation by gliricidia [28].

NFRE followed the same pattern observed for the Npdf values, where higher NFRE by corn was observed under 1000 plants ha^−1^ compared to 667 plants ha^−1^ (*p* < 0.05), with no differences between the other densities. The density of 1000 plants ha^−1^ was possibly a more favorable environment for Npdf absorption and consequently resulted in higher NFRE in corn grains. The lowest NFRE by corn grains, 29% and 9.7%, in the middle of the alley and adjacent to the trees, respectively, were obtained at the density of 667 plants of gliricidia ha^−1^. A study that was carried out in this same experimental area showed that the lower gliricidia density (667 plants ha^−1^) had lower increases in soil organic matter contents over the years [31] compared to the other tree densities, and this may be explained by the higher N losses of the N mineral fertilizer from the system. In a system that exclusively used gliricidia, the increase in plant density improved the soil chemical quality due to higher tree biomass residues and nutrient availability [7]. 

NFRE in corn grains was lower when the corn row was close to the gliricidia for all the tree densities evaluated. The lowest accumulated NFRE in corn grains was 9.74% in the line adjacent to the trees at the density of 667 plants ha^−1^. On the other hand, the highest NFRE observed was 45.78% in the corn row at the middle of the alley and for the density of 1000 plants ha^−1^ (Table 1). The highest mean NFRE in corn grains was similar to the values obtained in an intercropped corn/grass system without the arboreal component, in an area adjacent to the present study with NFRE ranging from 46 to 48% [21]. In the present study, the lower values of Npdf and NFRE by corn (*p* < 0.05) cultivated close to the gliricidia plants can be attributed to competition between the crop and the gliricidia for N-fertilizer uptake. 

The density of 1000 plants ha^−1^ also provided higher Npds content in corn grains (27.03 kg N ha^−1^) compared to 667 plants ha^−1^ (16.50 kg N ha^−1^). The accumulation of inorganic N in soil with gliricidia intercropped with corn was higher when compared to corn monocropping, as reported by Ikerra et al. [29] under tropical conditions (Southern Malawi) and Haplustalf soil (USDA classification). In systems where gliricidia is solely cultivated, it is known that the increase in tree density also increases soil fertility levels, mainly above 1000 plants ha^−1^ [7]. In our study, the main source of N for corn was the mineral fertilizer (Npdf), although other studies show different results. In an integrated system, the Npds was about 60% of the N contained in corn grains [21]. Possibly, the agrisilvicultural system with intercropping between gliricidia, corn and forage species reduced the Npdf losses in the soil, favoring the absorption of mineral N by corn compared to Npds.

The Npds was significantly reduced when the corn was closer to the trees of gliricidia (*p* < 0.05) compared to the corn rows at the middle of the alley for the plots at the density of 1333 plants ha^−1^ (Table 1). At the densities of 667 and 1000 plants ha^−1^, the proximity effect of gliricidia was not significant (*p* < 0.05) for the Npds in corn grains. Possibly, the limiting factor that decreased the Npds in the higher tree density group and in corn rows closer to the trees was the shading, which reduces corn growth [28]. 

There were no simple or interaction (*p* > 0.05) effects between plant density and the position of corn rows for Npds in corn straw (Table 1). Corn straw did not reflect the differences observed for the corn grains.

In general, corn rows that were closer to the plants of gliricidia (about 0.45 m distance) showed a competitive effect, and the rows at the center, at the middle of the alleys, were more efficient at recovering the N-fertilizer by the corn. The reductions observed for Npdf and Npds in the positions adjacent to the gliricidia were mainly due to competition for water, light and nutrients for all the gliricidia densities evaluated.

### 2.2. Nitrogen Use by P. maximum cv. Massai Intercropped with Corn in an Agrisilvicultural System with G. sepium

Differently from what was observed for corn, the tree densities and the position of corn rows did not affect (*p* > 0.05) the Npdf or the NFRE from the *P. maximum* shoots (Table 2). The Npdf in the forage shoots ranged from 1.69 kg N ha^−1^ to 5.45 kg N ha^−1^. The lack of difference in Npdf content in the shoots of *P. maximum* reinforces the low competition for Npdf between forage and corn as demonstrated in a study carried out in an area close to that of the present study, but without the presence of the tree component [21] and where the Npdf in forage shoots was, on average, 3.71 kg ha^−1^.

The forage shoots absorbed from 1.61% of the applied N-fertilizer to the corn in the position adjacent to the tree and within the system with 1333 plants ha^−1^ to 5.19% in the middle of the alley within the system with 1000 plants ha^−1^. These results demonstrate that the intercropping of tropical grass forages with corn in an agrisilvicultural system with gliricidia does not cause significant competition by the forage for the N-fertilizer applied to corn.

For the Npds in the shoots of *P. maximum*, there were significant interactions between tree densities and the position of corn rows. The highest values of Npds in forage (*p* < 0.05) were obtained at 1000 plants ha^−1^. The Npds in the *P. maximum* shoots ranged from 6.33 kg N ha^−1^ within the system with 1333 plants ha^−1^ to 16.22 kg N ha^−1^ within the 1000 plants ha^−1^ in the middle of the alley (Table 2). These values are very close to those obtained from shoots of *P. maximum* cv. Aruana and *B. humidicola* intercropped with corn, which ranged from 6.41 to 11.2 kg N ha^−1^ [21]. The density of 1000 plants ha^−1^, considering sampling at the middle of the alley, indicates greater synergy between the different productive components of the system (gliricidia, corn and grass), resulting in better quality of nutrition for the associated crops. Unlike corn, the main source of N for forage was the soil, as demonstrated by the higher Npds compared to the Npdf in forage shoots. There was no simple effect of position of corn rows on the Npds in the shoots of *P. maximum* (Table 2).

### 2.3. Nitrogen Use by the Crop Components of the Agrisilvicultural System (SYS) with G. sepium, Corn and P. maximum

There were significant interactions (*p* < 0.05) between the tree density and the position of corn rows for Npdf and NFRE considering the N accumulated in the crop components of the system (Table 3). On the other hand, the Npds that accumulated in the crop components of the agrisilvicultural system (straw and corn grain + grass shoots) were not affected (*p* > 0.05) by the different densities of gliricidia or by the position of corn rows in relation to the trees (Table 3).

The Npdf in corn (straw + grain) and *P. maximum* shoots in the agrisilvicultural system (Npdf-SYS) were higher (*p* < 0.05) in the density of 1000 plants ha^−1^ than with 667 plants ha^−1^, with no differences observed between the other densities (Table 3). When considering sampling at the middle of the alley, the density of 1000 plants ha^−1^ stood out as a favorable arrangement for the best use of N by the agrisilvicultural system.

There was only the effect of the sampling position on Npdf-SYS for the density of 1000 plants ha^−1^, resulting in higher (*p* < 0.05) Npdf in plants located in the central position of the plot. The lower competition in the middle of the alley and within the density of 1000 plants ha^−1^ was possibly the reason for the higher values of Npdf-SYS, as observed by Schroth and Zech [32]. Again, the pruning management before and during the cultivation of crops associated with gliricidia should be adopted to reduce competition in the agrisilvicultural system.

The recovery efficiency of the N-fertilizer by the agrisilvicultural system (NFRE-SYS) was higher (*p* < 0.05) with 1000 plants ha^−1^ (89%) compared to the system with 667 plants ha^−1^ (60.3%) in the central position of the plot (Table 3). Similar to the Npdf-SYS, at 1000 plants ha^−1^, the NFRE was higher in plants located in the middle of the plot. In addition to the shading effects and the possible competition for soil and water [2,32], the results show that gliricidia may have absorbed N-fertilizer from the corn rows closer to the trees. This has been addressed in a Typic Kandiudults in Indonesia [16], in which the authors estimated that 44 to 58% of the N in gliricidia was from biological N fixation, whereas the rest (42 to 56%) was absorbed from the available N in soil. Similarly, gliricidia absorbed substantial amounts of ^15^N-fertilizer during cultivation cycles, recovering about 11.5% of N applied as urea after 180 days of its application [26]. It has been highlighted that when gliricidia is cultivated exclusively, at densities greater than 1000 plants ha^−1^, there is competition between plants for water, light and nutrients [7]. Possibly, the competition for photosynthetic radiation between gliricidia and the crops with increasing planting density [7] has resulted in the intermediate NFRE-SYS values obtained in the system with 1333 plants ha^−1^. 

There was no effect of the density of gliricidia plants or the proximity of the trees on the N content derived from the soil in the agrisilvicultural system (Npds-SYS) accumulated in the corn and grass. Npds-SYS ranged from 35.99 kg ha^−1^ in the position adjacent to the tree at the density of 667 gliricidia plants ha^−1^ to 62.69 kg ha^−1^ in the middle of the alley at 1000 plants ha^−1^. Higher Npds have been obtained in intercropped systems when compared to monocropping with and without the tree component [21,33].

### 2.4. Return of N to the Soil from the Plant-Derived N from Fertilizer and Soil in an Agrisilvicultuiral System with G. sepium

The Npdf that returned to the soil through corn straw and shoots of the forage *P. maximum* was higher in the density of 1000 plants ha^−1^ in the position adjacent to the gliricidia plants, returning 12 kg ha^−1^ more N than at the density of 667 plants ha^−1^ (Figure 1). This indicates that competition for light was the most relevant factor because the plants recovered the N-fertilizer well, without being converted into grains. In the system with higher tree density, the Npdf that returned to the soil was intermediate between the other densities. In all cases, corn straw was the component that promoted the highest return of Npdf to the soil, representing more than 90% of the total, resulting from the higher N-fertilizer uptake by corn.

At the density of 667 plants of gliricidia ha^−1^, the Npdf that returned to the soil was similar between the positions of corn rows, being 31.08 kg N ha^−1^ in line A and 32.91 kg N ha^−1^ of Npdf in line B. At the density of 1000 plants ha^−1^, the agrisilvicultural system returned 45.38 kg ha^−1^ in line A and 37.15 kg ha^−1^ in line B. At the density of 1333 plants ha^−1^, the system returned 40.5 and 39.26 kg ha^−1^ of the Npdf in lines A and B, respectively. These results prove that corn is the major contributor of N to the soil in the agrisilvicultural system due to its high demand for N for grain production. In addition, the arrangement with 1000 gliricidia plants ha^−1^ was more efficient, contributing to a greater accumulation of N in the plant and soil. 

The Npds that returned to the soil by the shoots of *P. maximum* and corn straws ranged from 25.99 kg ha^−1^ in the system with 1333 plants ha^−1^ to 35.66 kg ha^−1^ at 1000 plants ha^−1^ (Figure 2). Corn straw contributed more to N returned to the soil than the shoots of the grass. Nevertheless, the contribution of grass was higher in Npds than in the Npdf.

### 2.5. Soil Total (STN) and Available Nitrogen (SAN) in Agrisilvicultural System with Different Population Densities of G. sepium

There was no significant effect of gliricidia plant density and the sampling position on soil total N (STN) content (Table 4). This lack of effect for STN possibly reflected the short period of establishment and management of the agrisilvicultural system and the rapid mineralization of plant biomass (e.g., leaves fallen from the tree legume). It was expected that the gains of SOM previously verified in the same experimental area [31] would be reflected in increments for STN. Other authors also indicated gains in SOM and carbon due to the intercropped gliricidia–corn system [6]. In a previous study in the same experimental area, the agrisilvicultural system with *P. maximum* cv. Massai and *G. sepium* managed to accumulate 13.9 Mg ha^−1^ of organic C in the 0.0–0.4 m soil layer over a period of 4 years [31]. Similarly, livestock systems with live fences of *G. sepium* (distanced up to 6 m from the pasture) increased SOM, carbon and N in the 0–0.1 m soil layer [34].

Tree density and sampling position factors showed a significant interaction, considering the soil available N (SAN). The higher density of gliricidia plants showed the highest values for SAN (34.90 mg kg^−1^) in the position adjacent to the plants, probably due to the higher deposition of fallen leaves and the greater accumulation of SOM fractions over the years [6,31]. The highest values of SAN in the presence of gliricidia have been explained as follows: (i) the N biologically fixed by gliricidia can be made available for corn through pruning and its incorporation into the soil, and (ii) by the greater capacity of gliricidia to absorb N at deeper soil layers and prevent it from being leached [3] or lost to the atmosphere [23,24].

Under 1333 plants of gliricidia ha^−1^, the SAN was higher in the sampling position adjacent to the gliricidia plants, while in the other tree densities the opposite was observed—higher SAN contents in the middle of the plot/alley. SAN contents, in line B, were not affected by the tree density, demonstrating little or no influence of the density of gliricidia plants on SAN contents in this sampling position. The higher availability of N in the soil may have resulted from the greater supply capacity of N-rich organic material and greater root exudation of N in the position adjacent to the trees [27].

### 2.6. Corn and Grass Yields in Agrisilvicultural Systems with Different Densities of G. sepium

There was no significant effect (*p* > 0.05) of the interaction of gliricidia plant densities and sampling position on corn grain and straw yields and the production of shoots of *P. maximum* (Table 5). Nevertheless, in all tree densities, corn grain and straw yields were higher in line B compared to line A. This demonstrates that the proximity of gliricidia to corn rows reduced corn yield through competition for photosynthetic radiation, water and nutrients. Schroth and Zech [32] also reported higher corn yield in the central line of alleys, compared to corn monocropping, as a result of the competition with gliricidia. Again, the results from the present study reinforce the need to frequently prune the gliricidia to reduce the interference in corn yield, as observed in previous studies [2,9].

Unlike the results observed for corn, the shoots of *P. maximum* did not have their yields limited by the sampling position adjacent to the gliricidia plants. The forage *P. maximum* cv. Massai was shown to be tolerant to shading and to competition for photosynthetic radiation, water and nutrients, demonstrating high biomass production potential in the position adjacent to the gliricidia plants.

## 3. Materials and Methods

### 3.1. Experimental Area

The experiment was established at the Experimental Farm Agua Limpa, University of Brasilia, Federal District, Brazil (latitude of 15°56′ S, the longitude of 47°56′ W, and altitude of 1090 m). The climate in the region is classified as Tropical Wet—Cwa, according to the Köppen classification, with an average annual rainfall of 1439 mm (concentrated between October and March) and mean annual temperatures varying from 16.7 °C to 22.4 °C.

All treatments were installed in a clayey Oxisol (Typic Haplustox) (Soil Survey Staff 2014), Latossolo Vermelho, according to the Brazilian Soil Classification System [35] or Gibbsic Ferralsol [36]. Chemical and physical attributes of the soil before establishing the trial are displayed in Appendix A.

### 3.2. History and Experimental Design

In 2012, the experimental area (approximately one hectare) was under low-productivity pasture. In December 2012, to establish the agrisilviculture system, the area was limed to raise base saturation to 50% (dolomite lime, 1.5 t ha^−1^), fertilized (87 kg ha^−1^ de P_2_O_5_), ploughed and harrowed up to the 0.00–0.20 m layer. The corn was sown (as no-tillage) in January 2013 with rows spaced 0.9 m apart, totaling approximately 60,000 plants ha^−1^. The corn was fertilized according to crop-specific requirements based on soil chemical analysis [37]. In January 2013, the cover crop—perennial grass—*Panicum maximum* cv. Massai was broadcasted at 10 kg ha^−1^ (considering the pure live seed percentage) one day after corn sowing. A description of the seasonal crop management and fertilization during the experimental study period is shown in Appendix A. Appendix A shows the sequence of operations performed in the experimental area over the years. Gliricidia was planted in January 2015. 

The experiment was arranged in a randomized complete block design with three replications. The following treatments were studied: (1) 667 plants of gliricidia ha^−1^, (2) 1000 plants of gliricidia ha^−1^ and (3) 1333 plants of gliricidia ha^−1^. The spacing between the gliricidia lines was 5 m and between plants it was 3.0 m (667 plants ha^−1^), 2.0 m (1000 plants ha^−1^) and 1.5 m (1333 plants ha^−1^). These plant densities were stipulated to test the effect of the largest and the smallest density of plants, having as a reference the spacing normally adopted for the exclusive planting of gliricidia for the region. 

The experimental plots had an area of 100 m^2^ (20 m × 5 m). Between the 5 m alleys of gliricidia, there were five lines of corn, spaced 0.9 m from each other. In December 2016, a drastic pruning was performed on the gliricidia plants, leaving only the stem with a height of 1 m. Gliricidia biomass (a mix of stem and leaf) obtained after pruning in each experimental plot was weighed, ground and evenly distributed in the experimental plots in December 2016. The average weight of fresh biomass derived from pruning per treatment is shown in Appendix A. 

### 3.3. Soil and Plant Sampling and Sample Preparation

Soil samples were collected in March 2017, at corn flowering, at a depth of 0–0.2 m. Each sample consisted of six sub-samples (two in the line and four in between the corn line). Each sample was collected spatially in the first corn row adjacent to the tree legume (A), and in the corn row in the center of the plot, middle of the gliricidia alleys (B), as shown in Appendix A. Soil samples were collected at 0.45 m and 2.50 m away from the gliricidia lines, represented in Appendix A by A and B, respectively. 

The agronomic yield variables studied were (1) corn grain yield, (2) aerial biomass of corn (without grain, and referred to here as straw), and (3) shoots of forage *P. maximum* cv. Massai. Corn and forage sampling were carried out in two positions in the plot: (A) in the position adjacent to gliricidia (0.45 m away from the gliricidia line) and (B) central position of the plot (2.5 m away from the gliricidia line). Corn for yield and straw production was sampled by collecting ten and five plants in the corn rows, respectively. To obtain the dry matter yield of shoots of *P. maximum*, in July 2017, two samples were collected using a square of 0.5 m^2^ in each position (A and B; Appendix A) in the experimental plot. These samples (corn grains and straw, and forage grass shoots) were oven-dried at 65 °C for 72 h to obtain the dry mass. The dried samples were finely ground in a Willey mill and used to determine total N by the Kjeldahl method, and the isotopic composition of the ^15^N.

### 3.4. Determination of Total N in Soil

Soil total N (TN) was determined by dry combustion in an elemental analyzer CHN (model EA 3000, CNH/O, Eurovector). Approximately 5 mg of macerated and sieved (0.149 mm mesh sieve) soil was used for analysis.

### 3.5. Analysis of Available N in Soil

The plant-available mineral nitrogen (nitrate + ammonium), henceforth referred to as soil available N (SAN), was determined by the extraction method of Na_3_PO_4_/borax buffer (pH 11.2) + NO_3_^−^ [21]. Calculations were performed from the calibration curve obtained by the distillation of standard solutions containing N (0, 15, 30, 45 and 60 mg N mL^−1^). The extracted N was determined by colorimetric spectrophotometry at 440 nm.

### 3.6. Study with ^15^N-Enriched Fertilizer

To study the dynamics of N from the ^15^N-enriched fertilizer, microplots of 1.5 m long and 0.9 m wide were allocated in the corn rows adjacent to trees (A) and in the corn rows in the middle of the plots (B) between the gliricidia alleys (Appendix A). Corn was planted with 350 kg ha^−1^ of fertilizer NPK 4-30-16, providing 14 kg N ha^−1^, 105 kg P_2_O_5_ ha^−1^ and 56 kg K_2_O ha^−1^. Sidedressing application was divided into two applications, where the first was performed at the V4 corn growth stage with the application of 52 kg N ha^−1^ (0.8% ^15^N in excess) and 60 kg K_2_O ha^−1^, and the second with a further 52 kg N ha^−1^ (0.8% ^15^N in excess), totaling 104 kg N ha^−1^ in the form of ammonium sulfate. At full corn flowering, flag leaf samples were collected in the two studied positions (A and B; Appendix A) per experimental plot. In July 2017, at corn harvest, corn grain and straw samples and forage shoot samples were collected within the useful area of each microplot (0.9 m^2^). The isotopic composition of the ^15^N was analyzed in a mass spectrometer (Thermo Finnigan Delta Plus; Thermo Fisher Scientific, Waltham, MA, USA) at Embrapa Agrobiology, Seropédica, Rio de Janeiro, Brazil.

The equations used to calculate N uptake and the N derived from fertilizer or soil, and N-fertilizer use efficiency (EUN) by corn or by grass have been previously described [38]. Nitrogen in the plant derived from fertilizer (Npdf) was calculated according to Equation (1).
(1)$$\mathrm{Npdf}={(}^{15}\mathrm{Npl}{/}^{15}\mathrm{Nf})\times \;\mathrm{tN}$$
where Npdf is in kg ha^−1^, ^15^Npl represents atoms of ^15^N in plant (straw or grain; in %), ^15^Nf represents atoms of ^15^N in fertilizer (%) and tN is total N accumulated in dry matter of corn or grass in kg ha^−1^.

N in plants derived from the soil was considered as all N taken up by corn or by grass not derived from the labelled mineral fertilizer, according to Equation (2).
(2)$$\mathrm{Npds}=\left(1-\mathrm{Npdf}\right)\times \;\mathrm{tN}$$
where Npds is the N derived from the soil (kg ha^−1^), Npdf is the relationship atom%^15^Npl/atom%^15^Nf, and tN represents total N in dry matter of corn or grass (kg ha^−1^).

The N-fertilizer recovery efficiency (NFRE) by corn was calculated according to Equation (3).
(3)$$\mathrm{NFRE}=\left(\mathrm{Npdf}/\mathrm{Af}\right)\times 100$$
where Npdf in kg ha^−1^ and Af is the total amount of the applied fertilizer (104 kg N ha^−1^).

The use of N was also evaluated for the cropping system (SYS) which was considered the sum of N in corn (grains + straw) and grass (shoots).

### 3.7. Statistical Analysis

For statistical analysis, a 3 × 2 factorial scheme was adopted, with three replications, considering three planting densities of gliricidia and two positions in the plot. The data were subjected to analysis of variance using THE GLM PROC from the SAS System for Windows software [39] and the means were compared by Tukey’s Studentized range test (*p* < 0.05).

## 4. Conclusions

This study demonstrates that the N-fertilizer use efficiency was influenced by the densities of gliricidia plants and by the sampling position of corn rows within the alley of the trees. The 1000 plants ha^−1^ density (compared to the 667 plants ha^−1^) was more efficient for the Npdf used by corn, forage and the agrisilvicultural system as a whole. The agrisilvicultural system showed a high recovery efficiency of N-fertilizer, reaching a value of 89% in the density of 1000 ha^−1^ of gliricidia plants. The corn that was close to the gliricidia plants competed with them, reducing the nitrogen fertilizer recovery efficiency by corn. In contrast to several studies that evaluate the N use efficiency by corn in monocropping, the integrated system showed that the largest source of N for corn was the N-fertilizer applied (and not the N derived from the soil), demonstrating that this system is capable of preserving the N from mineral fertilizer for corn, reducing the losses of this nutrient. In general, grain and corn straw yields were reduced in the position adjacent to the gliricidia plants. Therefore, the density of 1000 plants of gliricidia ha^−1^ is recommended for obtaining higher nitrogen fertilizer recovery efficiency and yields for the intercropped crops (corn and grass) in an agrisilvicultural system. As a perspective, further studies should consider a longer period of evaluation and a deeper soil layer must be assessed.

## Figures and Tables

**Figure 1 plants-12-01647-f001:**
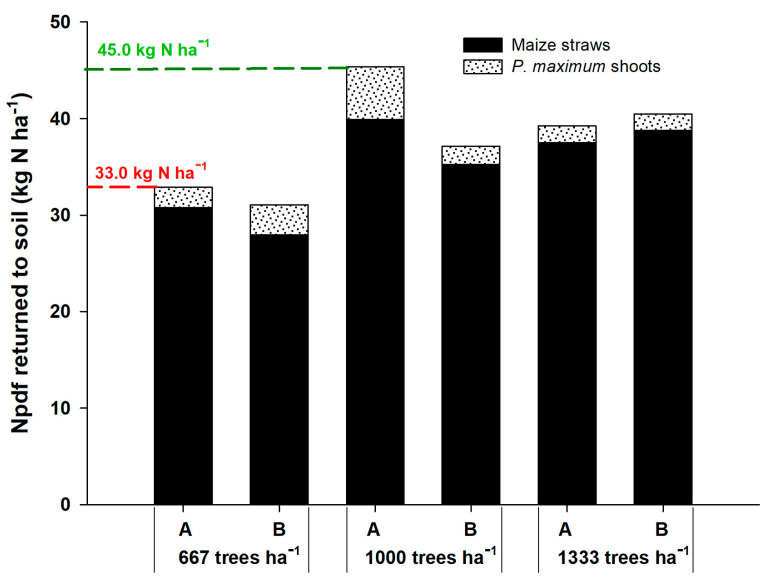
Nitrogen in plant derived from fertilizer (Npdf) that returned to the soil by corn straws and shoots of *P. maximum* in agrisilvicultural systems with 667, 1000 and 1333 plants of *G. sepium* ha^−1^. A and B represent the corn rows adjacent to the trees and in the center of the plots, respectively.

**Figure 2 plants-12-01647-f002:**
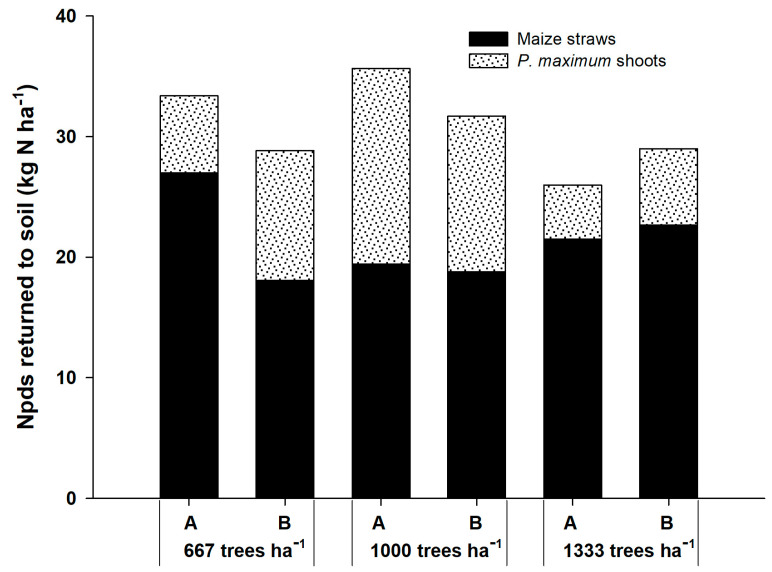
Nitrogen in plant derived from the soil (Npds) that returned to the soils by corn straws and shoots of *P. maximum* in three agrisilvicultural systems with 667, 1000 and 1333 plants of *G. sepium* ha^−1^. A and B represent the corn rows adjacent to the trees and in the center of the plots, respectively.

**Table 1 plants-12-01647-t001:** Nitrogen in plant derived from fertilizer (Npdf), N fertilizer recovery efficiency (NFRE) and N in plant derived from soil (Npds) in corn grains and straws under agrisilvicultural system with 667, 1000 and 1333 plants of *Gliricidia sepium* ha^−1^. Corn plants were collected at lines close to the *G. sepium* plants (A) and in the middle of the plots—middle of the alleys (B).

Npdf (kg N ha^−1^)						
Corn grains						
Line	667 plants ha^−1^		1000 plants ha^−1^		1333 plants ha^−1^	
A	30.46	bA	48.07	aA	44.39	abA
B	10.23	bB	31.43	aB	18.60	abB
Corn Straw						
	667 plants ha^−1^		1000 plants ha^−1^		1333 plants ha^−1^	
A	30.77	aA	39.93	aA	37.52	aA
B	27.98	aA	35.26	aA	38.81	aA
NFRE (%)						
Corn grains						
Line	667 plants ha^−1^		1000 plants ha^−1^		1333 plants ha^−1^	
A	29.01	bA	45.78	aA	42.27	abA
B	9.74	bB	29.93	aB	17.71	abB
Corn Straw						
	667 plants ha^−1^		1000 plants ha^−1^		1333 plants ha^−1^	
A	29.30	aA	38.03	aA	35.74	aA
B	33.44	aA	33.58	aA	36.96	aA
Npds (kg N ha^−1^)						
Corn grains						
Line	667 plants ha^−1^		1000 plants ha^−1^		1333 plants ha^−1^	
A	16.50	bA	27.03	aA	26.42	abA
B	7.15	bA	19.18	aA	11.56	abB
Corn Straw						
	667 plants ha^−1^		1000 plants ha^−1^		1333 plants ha^−1^	
A	27.00	aA	19.45	aA	21.52	aA
B	18.07	aA	18.79	aA	22.67	aA

Within columns (capital letters), lines (low case letters) and parameters, means followed by the same letter are not significantly different by the Tukey test (*p* < 0.05).

**Table 2 plants-12-01647-t002:** Nitrogen in the plant derived from fertilizer (Npdf), N fertilizer recovery efficiency (NFRE) and N in the plant derived from soil (Npds) in shoots of *P. maximum* cv. Massai from three agrisilviculture systems with 667, 1000 and 1333 plants of *Gliricidia sepium* ha^−1^. *Panicum maximum* shoots were collected at lines close to the *G. sepium* plants (A) and in the middle of the plots (B).

Npdf (kg N ha^−1^)						
*P. maximum* shoots						
Line	667 plants ha^−1^		1000 plants ha^−1^		1333 plants ha^−1^	
A	2.14	aA	5.45	aA	1.74	aA
B	3.10	aA	1.89	aA	1.69	aA
NFRE (%)						
*P. maximum* shoots						
Line	667 plants ha^−1^		1000 plants ha^−1^		1333 plants ha^−1^	
A	2.04	aA	5.19	aA	1.66	aA
B	2.95	aA	1.80	aA	1.61	aA
Npds (kg N ha^−1^)						
*P. maximum* shoots						
Line	667 plants ha^−1^		1000 plants ha^−1^		1333 plants ha^−1^	
A	6.39	bA	16.22	aA	4.47	bA
B	10.77	aA	12.91	aA	6.33	aA

Within columns (capital letters), lines (low case letters) and parameters, means followed by the same letter are not significantly different by the Tukey test at *p* = 0.05.

**Table 3 plants-12-01647-t003:** Total nitrogen in the plant derived from fertilizer (Npdf-SYS), N fertilizer recovery efficiency (NFRE-SYS) and N in the plant derived from soil (Npds-SYS) in corn grains, straws and in *P. maximum* shoots representing the amount of N accumulated in agrisilviculture systems with 667, 1000 and 1333 plants of *Gliricidia sepium* ha^−1^. Corn straws and grains and *Panicum maximum* shoots were collected at lines close to the *G. sepium* plants (A) and in the middle of the plots (B).

Npdf-SYS (kg N ha^−1^)						
Corn grains + straws + *P. maximum* shoots						
Line	667 plants ha^−1^		1000 plants ha^−1^		1333 plants ha^−1^	
A	63.37	aA	93.45	bA	83.66	abA
B	41.30	aA	68.58	aB	59.09	aA
NFRE-SYS (%)						
Corn grains + straws + *P. maximum* shoots						
Line	667 plants ha^−1^		1000 plants ha^−1^		1333 plants ha^−1^	
A	60.35	aA	89.00	bA	79.67	abA
B	39.33	aA	65.31	aB	56.28	aA
Npds-SYS (kg N ha^−1^)						
Corn grains + straws + *P. maximum* shoots						
Line	667 plants ha^−1^		1000 plants ha^−1^		1333 plants ha^−1^	
A	49.89	aA	62.69	aA	52,41	aA
B	35.99	aA	50.88	aA	40.56	aA

Within columns (capital letters), lines (low case letters) and parameters, means followed by the same letter are not significantly different by the Tukey test at *p* = 0.05.

**Table 4 plants-12-01647-t004:** Soil total (STN) and available nitrogen (SAN) from agrisilviculture systems with 667, 1000 and 1333 plants of *Gliricidia sepium* ha^−1^. Soil samples were collected at lines close to the *G. sepium* plants (A) and in the middle of the plots (B) and at the 0.00–0.20 m soil layer.

**STN (g kg ^−1^)**						
Line	667 plants ha^−1^		1000 plants ha^−1^		1333 plants ha^−1^	
A	1.82		1.85		1.75	
B	1.81		1.81		1.92	
**SAN (mg kg ^−1^)**						
Line	667 plants ha^−1^		1000 plants ha^−1^		1333 plants ha^−1^	
A	26.66 aA		28.56 aA		25.87 aB	
B	19.80 bB		18.53 bB		34.90 aA	

Within columns (capital letters), lines (low case letters) and parameters, means followed by the same letter are not significantly different by the Tukey test at *p* = 0.05.

**Table 5 plants-12-01647-t005:** Corn grain yield and above-ground biomass of corn and *P. maximum* cv. Massai from agrisilviculture systems with 667, 1000 and 1333 plants of *G. sepium* ha^−1^. Corn straws, grains and *P. maximum* shoots were collected at lines close to the *G. sepium* plants (A) and in the middle of the plots (B).

Line	Density of *G. sepium* Plants					
	667 plants ha^−1^		1000 plants ha^−1^		1333 plants ha^−1^	
	Corn grain yield (kg ha^−1^)					
B	5958	±800 a	7465	±1372 a	6167	±633 a
A	4221	±600 b	4960	±1366 b	4610	±700 b
	Corn above-ground biomass (kg ha^−1^)					
B	4095	±377 a	4475	±1331 a	4443	±239 a
A	3398	±417 b	3426	±697 b	3505	±672 b
	*P. maximum* above-ground biomass (kg ha^−1^)					
B	4590	±347 a	4187	±1031 a	3790	±439 a
A	3610	±457 a	4000	±697 a	4190	±972 a

Within lines (low case letters) and parameters, means followed by the same letter are not significantly different by the Tukey test at *p* = 0.05.

## Data Availability

The data presented in this study are available on request from the corresponding author.

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
