# Peer review of "Nitrogen Use Efficiency in an Agrisilviculture System with Gliricidia sepium in the Cerrado Region"

_plants, 2023, doi:10.3390/plants12081647_

Round 1

Reviewer 1 Report

Abstract

Line 20 please ha-1 changed to ha-1 as well please check it all over the manuscript.

Line 22 representing an excellent revise to representing as an excellent………...

Results

Line 190 please revise Kg ha-1 to Kg ha-1

Line 300 in Figure 2 caption, 667, 1000, e 1333 what does “e” means I observed it in many places please made it clear. 

Line 350 P. maximum is not in italic form while in many places it’s in an italic form please keep the same format throughout the manuscript. 

Materials and Methods 

Line 362 in the experimental area there is a dot (.), I don’t think so to put a dot after the subtitle. in the previous sub-topic, there is no dot as well you put. 

If the co-authors write the quantity of nitrogen in the soil would be much better. 

Major revisions

The study is quite interesting but I don’t think so to Jude the performance of nitrogen by single year field trial. In an open environment, the conditions could be changed at any time. Therefore, to observe the performance of nitrogen for the current research I suggest the co-authors do the experiment twice. 

Author Response

Reviewer #1

Abstract

Reviewer: Line 20 please ha-1 changed to ha-1 as well please check it all over the manuscript.

Authors: It has been checked and corrected throughout the manuscript.

Line 22 representing an excellent revise to representing as an excellent...

Authors: It has been corrected.

Results

Reviewer: Line 190 please revise Kg ha-1 to Kg ha-1

Authors: It has been corrected.

Reviewer: Line 300 in Figure 2 caption, 667, 1000, e 1333 what does “e” means I observed it in many places please made it clear.

Authors: Thank you for the suggestion. It has been corrected throughout the text.

Reviewer: Line 350 P. maximum is not in italic form while in many places it’s in an italic form please keep the same format throughout the manuscript.

Authors: It has also been corrected throughout the text.

Materials and Methods

Reviewer: Line 362 in the experimental area there is a dot (.), I don’t think so to put a dot after the subtitle. in the previous sub-topic, there is no dot as well you put.

Authors: It has been corrected throughout the text. Dots have been deleted from the subtitles.

Reviewer: If the co-authors write the quantity of nitrogen in the soil would be much better.

Authors: Soil total nitrogen content has been added to Table S1.

Major revisions

The study is quite interesting but I don’t think so to Jude the performance of nitrogen by single year field trial. In an open environment, the conditions could be changed at any time. Therefore, to observe the performance of nitrogen for the current research I suggest the co-authors do the experiment twice.

Authors: We respect the reviewer's comment. However, there are hundreds of excellent papers (using the 15N isotope technique) published in respected journals with only one cultural cycle. In addition, 'Plants' has published works using the 15N technique in experiments whose evaluations were made in just one year. Our work was conducted over four years with a complex interaction involving maize, grass and trees. We firmly believe that our work is unique and presents unprecedented results on N cycling in integrated production systems. 

Reviewer 2 Report

The manuscript shows the results of a so-called agrisilviculture experiment, which is of interest for agriculture in the tropics. The approach and the first results are of interest. The validity of the results is somewhat limited by the short duration of the experiment and by limited data on microclimate and soil. Further comments to the manuscript are to find within the attachment. 

Author Response

Reviewer #2

Reviewer: The manuscript presents the results on N uptake and efficiency of maize and Panicum maximum under the conditions of an agrisilviculture system with different plant densities of the tree species Gliricidia sepium. The method of 15N isotope tracers is applied. The method used is innovative and reliable to precisely determine the N uptake of the plants. Overall, the question is of particular interest for agriculture in the tropics. The short duration of the experiment (short duration of the agroforestry system) and the one-year nature of the yield results are judged to be unfavourable.

Authors: We thank Reviewer 2 for his/her comments. As the same question was made by Reviewer 1, we are repeating our response as follows:

We respect the reviewer's comment. However, there are hundreds of good works (using the 15N isotope technique; including pot experiments) published in respected journals with only one cropping growing. In addition, 'Plants' has published works using the 15N technique in experiments whose evaluations were made in just one year. Our work was conducted over four years with a complex interaction involving maize, grass and trees. We firmly believe that our work is unique and presents unprecedented results on N cycling in integrated production systems.

Reviewer: I have the following comments on the present manuscript regarding content and form:

  1. The trial was started with the planting of Gliricidia sepium (Gs) in 2015. Already in Dec. 2016, the biomass of Gs was distributed on the soil and incorporated. And again, one year later, the measurements were already taken. These time intervals are very short and allow only limited statements about the agroforestry system, which should be established over several years. The effects on the soil are likely to change after further years. This circumstance should be considered and evaluated in the manuscript.

Authors: We agree with the Reviewer, “the effects on the soil are likely to change after further years” and following further prunings. In the present study, gliricidia was planted in January, 2015. Therefore, in January 2017, when 15N experiment started, gliricidia trees were already in the third cycle (from Jan 2015 to Jan 2017). In addition, in the previous year (2016) the effects of the different arrangements of the agrisilvicultural system with only one tree species (gliricidia) were already seen for N cycling.

Reviewer:

  1. Soil samples were taken up to 0.2 m. Thus, all statements in the manuscript refer only to this soil depth. However, the maize roots are likely to go deeper than 0.2 m. Vertical displacements of C and N are also possible. For this reason, it would have been better if a greater soil depth or more soil layers (e.g., 0-0.2 m, 0.2-0.4 m, 0.4-0.6 m) had been considered. Since this was not done, the aspects of rooting depth and vertical distribution of soil nutrients should be addressed in the discussion.

Authors: We appreciate the reviewer’s comments. We agree that soil samples could have been taken from deeper layers specially to explain better the effects of root development from the different species and competition. However, this was not done. We have discussed a little about root competition in lines 113-120 from another trial in Malawi that could support the reviewer’s comment. We also believe that if the focus of the study was to evaluate C and N stocks in soil and determine N leaching, this should have been measured in deeper soil layers. Local farmers mainly take the 0.0 – 0.2 m layer for determining soil chemical parameters. In addition, information about the soil parameters under this same experimental site and agrisilvicultural system (layers 0.0 – 0.1, 0.1 – 0.2 and 0.2 – 0.4 m) can be checked in Coser et al. 2018 (Short-term buildup of carbon from a low-productivity pastureland to an agrisilviculture system in the Brazilian savannah – Agricultural Systems - ScienceDirect).

Reviewer:

  1. It is not clear to the reader how the experiment went. For example: At the end of 2013, animals entered the area to graze. Why was this done and what did the animals graze on (Panicum max.)? In the years that followed, however, no animals came to the area. I miss an explanation of the purpose of this measure.

Authors: To make the integrated system closer to what is adopted by farmers, only in the 2012/2013 season, animals (sheep) were introduced in the experimental area with similar stocking rates and grazing time in all experimental plots.

The following sentence has been added to the Table S2:

To simulate the grazing in the integrated system, at the end of July, 2013, after the corn harvest, animals (sheep; stocking rate of 10 AU/ha) enter the experimental area to graze on P. maximum over 20 days.

In the following years (from August 2013 up to 2017) no animals entered the plots for grazing.

Reviewer:

  1. When exactly was Gliricidia planted (date)?

Authors: Gliricidia was planted in January, 2015. It has been added to both the ‘Material and methods’ section (lines 394-395) and Table S2.

Reviewer:

  1. In 2017, Panicum max was harvested to measure yields. When exactly (date) did this harvest take place?

Authors: Yield measures were performed in July, 2017. This information has been added to the text (line 423) and Table S2.

Reviewer:

  1. What does "forage was mowed" (Dec. 2016) mean? Which plant is meant by "forage" and why "mowed" in Dec. 2016?

Authors: P. maximum was cut down (=mowed) in December 2016 before the corn planting. It has been added to Table S2.

Reviewer:

  1. Pruning of Gliricidia: What were the stem and leaf proportions of this Gliricidia biomass now incorporated into the soil? Were the C/N contents of this biomass determined? The twigs are low in N, while the leaves contain more N. Therefore, this information could be of importance. If no data are available, then this aspect should be briefly mentioned/explained in the text

Authors: We agree with the Reviewer, this information is very important.

Unfortunately, we did not separate the parts of the gliricidia biomass (stem and leaves). We opted to join the parts of the pruning that is a common practice in our region. Also, we did not determine the C and N contents of the pruned parts.

We have added this information to the text (line 406).

Reviewer:

  1. In the discussion, some comparisons are made with other trials or authors (Ikerra et al, Rowe et al, Chirwa et al, 30, 31). Are the conditions in the cited literature comparable to your own conditions? If soil values are compared, they should also refer to 0-0.2 m soil depth. Whether this is the case or not should be noted in the discussion.

Authors: Authors: Thank you for the comment - – we have reviewed the papers to further understand the relationship with our own study. In this way we have made the following changes:

1) Ikerra et al., 1999 was taken from lines 115 – 116 and added to lines 174-175. We have also added information about soil depth and soil type;

2) For Rowe et al., 2005 [26] and Rowe et al., 2001 [16], we added further information in lines: 136-139; 264;

3) For Chirwa et al., 2003 [28], we added further information to lines 138-139 [28].

4) For [30] and [31], we have added further information in lines 123-124.

Reviewer:

  1. SAN (soil available N): Does this mean the plant-available mineral (inorganic) nitrogen (nitrate + ammonium)?

Authors: Exactly. A correct description has been added to the text (lines 433-434).

Reviewer:

  1. To better understand the competition between maize/Panicum max. and Gliricidia, climatic or microclimatic measurements on the trial plot (incl. light/PAR) would have been useful. Presumably, these measurements were not carried out. Perhaps one can still discuss in the discussion what the possible causes for the competition effects were (nutrients, water uptake, light/shading, root spread etc.).

Authors: Yes, microclimatic measurements were not collected from the trial. Therefore, we have discussed possible causes for the competition in lines 113-11 (shading [28] and root development competition [16]. We have added further discussions about the results on root distribution from other trials in lines 114 – 120.

Reviewer:

  1. The quote in line 582 is in Portuguese. An English translation (in brackets behind) would be useful here.

Authors: Thanks for this suggestion. An English translation has been provided.

Reviewer 3 Report

Work comments

The work is suitable for printing after minor corrections.

  Chapter 3 Material and Methods should be before Results and Discussion. It should be chapter 2. It is difficult to read the discussion of the results without knowing the methodology.

In table S1, the content of Nitrogen and Carbon in the soil is missing.

Is 65 degrees enough to dry samples for dry matter content? In scientific research, a temperature of 105 degrees is used.

No citation using formulas, page 12.

Too general conclusions.

  verse 87 - please remove the dot

Author Response

Reviewer #3:

Reviewer: The work is suitable for printing after minor corrections.

Authors: Thank you very much for your comments.

Reviewer: Chapter 3 Material and Methods should be before Results and Discussion. It should be chapter 2. It is difficult to read the discussion of the results without knowing the methodology.

Authors: We agree with the Reviewer. However, we followed the recommendation of the Plants editorial board and the available template. If the editor-in-chief allows it, we will make the suggested change.

Reviewer: In table S1, the content of Nitrogen and Carbon in the soil is missing.

Authors: Carbon and nitrogen contents have been provided in Table S1.

Reviewer: Is 65 degrees enough to dry samples for dry matter content? In scientific research, a temperature of 105 degrees is used.

Authors: Usually, 65 degrees is the recommended temperature for plant biomass to avoid the combustion of organic matter at higher temperatures.

Reviewer: No citation using formulas, page 12.

Authors: Please see the following sentence of the text that includes the citation:

The equations used to calculate N uptake and the N derived from fertilizer or soil, and N-fertilizer use efficiency (EUN) by corn or by grass have been previously described [39].

Reviewer: Too general conclusions.

Authors: We consider that specific conclusions are shown in this section. The following sentence has been added to the Conclusions:

As a perspective, further studies should consider a longer period of evaluation and a deeper soil layer must be assessed.

Reviewer:  verse 87 - please remove the dot

Authors: Dot has been removed.